# Diagnostic Superiority of Dual-Time Point [^18^F]FDG PET/CT to Differentiate Malignant from Benign Soft Tissue Tumors

**DOI:** 10.3390/diagnostics13203202

**Published:** 2023-10-13

**Authors:** Philippe d’Abadie, Olivier Gheysens, Renaud Lhommel, François Jamar, Thomas Kirchgesner, Filomena Mazzeo, Laurent Coubeau, Halil Yildiz, An-Katrien De Roo, Thomas Schubert

**Affiliations:** 1Department of Nuclear Medicine, Cliniques Universitaires Saint Luc-Institut Roi Albert II, Université Catholique de Louvain, 1200 Brussels, Belgium; olivier.gheysens@uclouvain.be (O.G.); renaud.lhommel@uclouvain.be (R.L.); francois.jamar@uclouvain.be (F.J.); 2Department of Radiology, Cliniques Universitaires Saint Luc-Institut Roi Albert II, Université Catholique de Louvain, 1200 Brussels, Belgium; thomas.kirchgesner@uclouvain.be; 3Department of Clinical Oncology, Cliniques Universitaires Saint Luc-Institut Roi Albert II, Université Catholique de Louvain, 1200 Brussels, Belgium; filomena.mazzeo@uclouvain.be; 4Department of Abdominal Surgery, Cliniques Universitaires Saint Luc-Institut Roi Albert II, Université Catholique de Louvain, 1200 Brussels, Belgium; laurent.coubeau@uclouvain.be; 5Department of Internal Medicine, Cliniques Universitaires Saint Luc-Institut Roi Albert II, Université Catholique de Louvain, 1200 Brussels, Belgium; halil.yildiz@uclouvain.be; 6Department of Pathology, Cliniques Universitaires Saint Luc-Institut Roi Albert II, Université Catholique de Louvain, 1200 Brussels, Belgium; an-katrien.deroo@uclouvain.be; 7Department of Orthopedic Surgery, Cliniques Universitaires Saint Luc-Institut Roi Albert II, Université Catholique de Louvain, 1200 Brussels, Belgium; thomas.schubert@uclouvain.be

**Keywords:** FDG PET/CT, dual-time point acquisition, soft tissue tumor, sarcoma

## Abstract

[^18^F]FDG PET/CT is used in the workup of indeterminate soft tissue tumors (STTs) but lacks accuracy in the detection of malignant STTs. The aim of this study is to evaluate whether dual-time point [^18^F]FDG PET/CT imaging (DTPI) can be useful in this indication. In this prospective study, [^18^F]FDG PET/CT imaging was performed 1 h (t1) and 3 h (t2) after injection. Tumor uptake (SUVmax) was calculated at each time point to define a retention index (RI) corresponding to the variation between t1 and t2 (%). Sixty-eight patients were included, representing 20 benign and 48 malignant tumors (including 40 sarcomas). The RI was significantly higher in malignant STTs than in benign STTs (median: +21.8% vs. −2%, *p* < 0.001). An RI of >14.3% predicted STT malignancy with a specificity (Sp) of 90% and a sensitivity (Se) of 69%. An SUVmax_t1_ of >4.5 was less accurate with an Sp of 80% and an Se of 60%. In a subgroup of tumors with at least mild [^18^F]FDG uptake (SUVmax ≥ 3; *n* = 46), the RI significantly outperformed the diagnostic accuracy of SUVmax (AUC: 0.88 vs. 0.68, *p* = 0.01). DTPI identifies malignant STT tumors with high specificity and outperforms the diagnostic accuracy of standard PET/CT.

## 1. Introduction

Soft tissue (ST) tumors represent a large and heterogeneous spectrum of tumors that develop in various extraskeletal structures, including connective tissues, muscles, fat, peripheral nerves, cartilage, tendons, blood vessels, etc. These tumors are characterized by histological, immunochemical, and genetic patterns that highlight specific clinical behaviors [1]. ST tumors can be classified as benign or malignant based on their local aggressiveness and metastatic potential. Malignant ST tumors are rare, with a 1:100 ratio compared with benign tumors, and correspond to only 1% of neoplasms [2,3]. The majority of malignant ST tumors are sarcomas, which include more than 50 subtypes. Depending on the histogenetic characteristics (histologic grade, chromosomal abnormalities, etc.), some subtypes can be very aggressive with a high metastatic rate and poor prognosis, requiring specific therapeutic approaches [4]. The high diversity, rarity, intrinsic complexity, and intratumoral histological heterogeneity of ST sarcomas pose significant challenges to confirm pathological diagnosis, with inaccuracies reaching 30% [5,6]. Furthermore, due to the similar appearance of benign and malignant soft tissue tumors, the risk of diagnostic error is high. In a series of 581 lesions secondarily reviewed by an expert pathologist, major diagnostic errors affecting patient management were found in 148 cases (25%), with 20 benign lesions being reclassified as malignant [7]. To optimize tumor characterization, combining additional analyses such as imaging with pathological diagnosis may be beneficial. This has been illustrated by Kuhn et al. who showed that one should ensure that there is concordance between imaging and the pathologic diagnosis, especially in the case of diagnostic dilemmas [3]. Imaging allows for the non-invasive assessment of intralesion heterogeneity and may guide biopsies toward the most metabolically active regions, hence leading to more representative biopsies. Therefore, optimal management requires a multidisciplinary approach in reference centers with shared clinical, imaging, and pathological expertise [1,8]. In the setting of an ST mass, accurate management and diagnosis are critical to promptly determine the best treatment options and optimize the prognosis for aggressive sarcomas. 

MRI is the imaging modality of choice to characterize the exact anatomic location, size, and extent of a tumor and evaluate the malignant potential of an ST mass. Some morphologic criteria help to predict tumor malignancy, but an accurate diagnosis remains difficult in many cases [3]. The workup is almost always completed by a biopsy for determining the histological subtype and deciding the best treatment options. A direct excisional biopsy can be performed for small superficial lesions (<3 cm) [8]. Currently, a thoraco-abdominal CT scan remains the minimum for the extension workup; however, many expert centers have shifted toward [^18^F]FDG-PET/CT in their practices for the vast majority of ST masses, with the exception of lesions of myxoid or paucicellular content. [^18^F]FDG-PET/CT is useful for discriminating ST tumors, guiding the biopsy in a heterogeneous mass, and detecting distant metastases [9,10,11]. [^18^F]FDG-PET/CT assesses tumor glycolytic activity, which can be measured and quantified through the maximum Standardized Uptake Value (SUVmax). This metabolic tumor intensity is positively correlated with tumor malignancy and helps to differentiate malignant from benign tumors. High-grade sarcomas exhibit high FDG uptake in contrast to benign and low-grade sarcomas [12,13]. Therefore, FDG PET/CT can be useful for guiding the biopsy in the most FDG-avid part of tumors to more accurately predict the nature of the tumor and its aggressiveness [8]. Nevertheless, [^18^F]FDG PET/CT has some limitations for the characterization of ST tumors. Low-grade sarcomas exhibit low FDG uptake, similar to benign tumors, resulting in false-negative results [12]. However, certain benign tumors may exhibit high [^18^F]FDG avidity, leading to false-positive findings [14]. In addition to SUVmax, other techniques are available for assessing tumor metabolism. Some previous in vivo and in vitro research has shown that dual-time point [^18^F]FDG PET/CT imaging (DTPI) can aid in distinguishing malignant tumors from benign tumors, thus enhancing the specificity of [^18^F]FDG PET/CT [15]. Comparing standard [^18^F]FDG acquisition performed at 1 h post intravenous injection (PI) and delayed acquisition (2 or 3 h PI), [^18^F]FDG uptake would increase in malignant tumors and decrease or remain unchanged in benign lesions. 

The purpose of this study is to evaluate the usefulness of the delayed acquisition of [^18^F]FDG PET/CT to better characterize soft tissue tumors and predict malignancy. We aimed to investigate whether DTPI can increase the diagnostic accuracy of [^18^F]FDG PET/CT in soft tissue tumors.

## 2. Materials and Methods

### 2.1. Population

This prospective study aimed to enroll any patient with the indication of [^18^F]FDG PET/CT for undeterminable or potentially aggressive ST tumors discussed during our multidisciplinary oncology team meeting. Our tertiary hospital is a reference center for sarcoma. Every referred patient’s file is discussed in a multidisciplinary setting comprising surgical oncologists, pathologists, radiation and medical oncologists, radiologists, and nuclear medicine specialists. For soft tissue tumors, the usual workup includes medical history investigations and clinical examination. MRI imaging is aimed at identifying criteria for aggressiveness/malignancy, such as large size, deep location, heterogeneous MRI signal, contrast enhancement, invasion of surrounding structures, and rapid growth [16,17,18]. For lesions meeting those criteria, [^18^F]FDG PET/CT was performed to evaluate the metabolic activity of the ST tumor, identify the best region of interest for a biopsy, and rule out distant metastases. Subsequently, a biopsy and/or surgical excision of the tumor was performed, and a histologic analysis confirmed the diagnosis. All patients meeting the criteria for [^18^F]FDG PET/CT were proposed to enter the study.

In total, 68 patients were enrolled between August 2021 and June 2023 after approval by our local ethics committee (2021/02AOU/328). 

### 2.2. [^18^F]FDG PET/CT Acquisitions and Analyses

[^18^F]FDG PET/CT was performed according to standard recommendations [19]. Briefly, the patients were fasted for at least 4 h and had their blood glucose levels confirmed to be inferior to 200 mg/dL prior to intravenous administration of 280–310 MBq [^18^F]FDG. Whole-body PET/CT imaging was performed on a Gemini TF (Philips Medical Systems, Cleveland, OH, USA). Tumors with a short axis inferior to 15 mm were excluded from this study due to underestimation of [^18^F]FDG uptake caused by the partial volume effect. A standard PET acquisition was acquired at 60 min postinjection (PI) (t1) with an acquisition time of 1 min 30 s per bed position. All images were acquired on a TOF-PET/CT (Philips Gemini TF64) with a time-resolution of 600 ps. Standard FDG iterative reconstruction (OSEM) was used, i.e., 4 × 4 × 4 mm^3^ reconstructed voxel size (matrix 144 × 144) with 3 iterations × 33 subsets. CT images were acquired with standard parameters, i.e., a tube voltage of 120 kV and an effective tube current of up to 100 mA. 

A second delayed acquisition (t2) was obtained on the tumor site at 180 min PI, with a longer PET acquisition (3.5 min per bed position) aimed at reducing noise caused by fluorine-18 decay. This second acquisition was performed on the same PET/CT system to avoid SUV variability due to the equipment. 

A board-certified nuclear medicine physician with 12 years of experience reviewed each exam and evaluated the tumor [^18^F]FDG uptake at t1 and t2 using SUVmax. Analyses were performed using MIM software version 7.1.3 (MIM Software Inc., Cleveland, OH, USA). With this software, we automatically defined the SUVmax of the tumor using a volume of interest. Retention Index (RI) corresponds to the difference in SUVmax between t1 and t2 and was calculated using the following formula:(1)$$Retention\;Index\;\left(\%\right)=\frac{SUVmax_{t2}-SUVmax_{t1}}{SUVmax_{t1}}\times 100\;$$

A positive change in RI corresponded to an increase in [^18^F]FDG uptake during the delayed acquisition, whereas a negative RI indicated a decrease in [^18^F]FDG uptake during the delayed acquisition.

A secondary analysis was conducted in tumors exhibiting at least a mild metabolism, defined as an SUVmax of ≥3. Different SUVmax cutoffs were tested, and only an SUVmax of ≥3 showed significant differences in AUC values between SUVmax and RI. 

### 2.3. Histopathological Analyses

Pathological diagnosis was performed in accordance with the 2020 WHO classification for soft tissue tumors [1]. At least one sarcoma expert pathologist confirmed the final diagnosis by performing multiples analyses, including morphological examinations, immunohistochemistry, and molecular tests. Tumors were classified into two categories: benign and malignant. Tumors with uncertain malignant potential were excluded from the analysis. 

### 2.4. Statistical Analysis

The statistical analysis was conducted with MedCalc Software version 20.218 (MedCalc Software Ltd., Ostend, Belgium). 

We compared SUVmax (t1) and RI in the benign and malignant tumor groups using a Mann–Whitney U test. The diagnostic performance of SUVmax and RI in distinguishing between malignant and benign tumors was evaluated using Receiver-Operating Characteristic (ROC) curves. The sensitivity (Se) and specificity (Sp) cutoffs were determined using the Youden index, which provides an equal balance between false-positive and false-negative values. The diagnostic accuracy of SUVmax and RI was measured through the Area Under the ROC Curve (AUC), and comparisons were conducted using the DeLong test [20]. The results were reported with medians and interquartile range (IQR). A *p*-value of <0.05 was considered statistically significant. 

## 3. Results

### 3.1. Patient and Tumor Characteristics

In total, 68 patients were included in this study (34 males and 34 females), with a median age of 62.5 years (IQR: 24). ST tumors involved the lower extremities in 35 patients (51.5%), the upper extremities in 17 patients (25%), the lower trunk/abdomen in 7 patients (10.2%), the upper trunk/chest in 6 patients (8.8%), and the neck in 3 patients (4.5%).

In total, 48 tumors (70.6%) were classified in the malignant category with a large proportion of sarcomas (40 tumors; 83.3%), and 20 tumors (29.4%) were classified as benign. 

One patient with a tumor of uncertain malignant potential (solitary fibrous tumor) was excluded from the analysis.

### 3.2. SUVmax and RI in the Different Tumor Subtypes

In malignant tumors, the median SUVmax was 6.8 (IQR: 12.6), and the median RI was +21.8% (IQR: 23.3%). 

In benign tumors, the median SUVmax was 2.7 (IQR: 2.3), and the median RI was −2% (IQR: 27.3%). 

Figure 1 illustrates a case of high-grade liposarcoma with high SUVmax and high positive RI. 

SUVmax and RI were significantly higher in malignant tumors than in benign tumors, as shown in Figure 2 and Figure 3 (*p*-values = 0.002 and <0.0001, respectively). 

The SUVmax and RI for each tumor subtype are shown in Table 1 for malignant tumors and in Table 2 for benign lesions/tumors. In the malignant category, some tumors in the soft tissues corresponded to bone sarcomas. 

In sarcomas, the median SUVmax was 7.0 (IQR: 13.1), and the median RI was +19.9% (IQR: 26.2%). As shown in Table 3, there were significant differences when comparing low-grade (grades 1–2) and high-grade sarcomas (grade 3). Low-grade sarcomas had low metabolic activity; the median SUVmax was 2.6 (IQR: 1.8), and the median RI was +0.4% (IQR: 32%). High-grade sarcomas had high metabolic activity, all with high SUVmax (median: 13.7 and IQR: 9.9) and high positive RI (median: +26.9% and IQR: 14.5%). 

### 3.3. Diagnostic Performance of DTPI in the Detection of Malignant ST Tumors

An SUVmax cutoff of >4.5 identified malignant tumors with an Se of 60.4%, Sp of 80%, and AUC of 0.74 [CI 95%: 0.62; 0.84]. Delayed SUVmax (t2) showed similar performance to SUVmax (t1) with a cutoff of >5.1 (Se = 60.4%, Sp = 85%). Their ROC curves were not significantly different (AUC = 0.76 [CI 95%: 0.64; 0.86], *p* = 0.06). 

An RI cutoff of >+14.3% identified malignant tumors with an Se of 69% and Sp of 90% (AUC = 0.82 [CI 95%: 0.70; 0.90]). When comparing SUVmax to RI, the AUCs were not significantly different (*p* = 0.17). 

In a subgroup of tumors with at least mild [^18^F]FDG uptake (SUVmax ≥ 3, *n* = 46), significantly higher performance was demonstrated using RI than using SUVmax (AUCs = 0.88 vs. 0.68; *p* = 0.01, Figure 4). For this subset of tumors, an RI of >17.9% identified malignant tumors with an Sp of 100% and an Se of 70.3%. 

## 4. Discussion

This original study demonstrates that DTPI can predict the malignancy of a soft tissue mass with high specificity by measuring the high retention of [^18^F]FDG uptake during delayed PET acquisition. In our data, [^18^F]FDG uptake on the delayed acquisition increased up to 68% on a malignant ST tumor (Table 1), and a high positive RI was a strong predictor of tumor malignancy (Sp: 90–100%). In contrast, the vast majority of benign ST tumors showed stable or decreased [^18^F]FDG uptake during delayed acquisition. These observations translate the glycolytic activity of tumor cells. Similar to glucose, [^18^F]FDG is accumulated in cells via GLUT transporters and is phosphorylated by a hexokinase enzyme during the first step of glycolysis. In contrast to glucose, [^18^F]-FDG-phosphate does not undergo further steps of glycolysis and accumulates in cells without the possibility of being released into the extracellular space. This accumulation of [^18^F]FDG in tumor cells is measured by PET/CT systems at the tissue level and is quantified by SUVmax [21,22]. The majority of malignant tumor cells exhibit high levels of GLUT transporters and hexokinase activity, and, therefore, [^18^F]FDG uptake at the tissue level is intense (i.e., high SUVmax). In addition, tumor cells with high hexokinase activity would demonstrate a significant increase in [^18^F]FDG uptake in a given time period and, therefore, a significant increase in [^18^F]FDG uptake during delayed PET acquisition (i.e., high positive RI). In contrast, benign tumor cells often exhibit low levels of GLUT transporters and low hexokinase activity, which are responsible for low [^18^F] FDG uptake. In addition, nonmalignant cells may have high levels of glucose-6-phosphatase, which is responsible for the release of [^18^F]FDG, therefore reducing uptake during delayed PET acquisition (i.e., negative RI) [23]. DTPI could be very useful in reducing the false-positive rate of [^18^F]FDG PET/CT in the characterization of ST tumors. For instance, four desmoid tumors exhibited high [^18^F]FDG uptake but no significant retention index (Table 2). In our data, DTPI was very efficient for distinguishing malignant tumors, especially those with at least mild initial [^18^F]FDG uptake (Figure 3, specificity reaching 100%). Nevertheless, this delayed PET acquisition is useless for tumors with low [^18^F]FDG uptake (tumors with an SUVmax of <3 in our study). Therefore, DTPI cannot help differentiate low-grade sarcomas from benign lesions, both of which often show mild [^18^F]FDG uptake. Independent of their metabolic activity, all lesions should still be biopsied to obtain a final histopathologic diagnosis. 

Our results are consistent with some previous studies. In a group of 29 patients with soft tissue masses who underwent PET acquisitions over a 6 h period, Lodge et al. demonstrated significant differences in time–activity uptake between benign tumors and high-grade sarcomas [24]. Furthermore, the usefulness of DTPI for malignant tumor detection has been demonstrated in several other tumor types [25,26,27,28,29,30]. 

DTPI did not significantly outperform conventional PET/CT in terms of sensitivity (Se 69% vs. 60%). A significant proportion of malignant ST tumors did not show high [^18^F]FDG uptake and significant RI. Interestingly, malignant tumors with low [^18^F]FDG uptake and/or without significant positive RI corresponded to low-grade tumors, such as low-grade sarcomas and a soft tissue chordoma metastasis (Table 1 and Table 3). Conversely, malignant tumors with high [^18^F]FDG uptake and/or high RI corresponded to aggressive malignant tumors, such as high-grade sarcomas (Table 1 and Table 3). Therefore, DTPI seems to be very useful for the detection of aggressive malignancies, such as high-grade ST sarcomas, and could help the pathologist classify uncertain ST tumors after biopsy [31]. ST sarcomas represent a very heterogeneous group of tumors with approximately 70 histologic subtypes [1]. Histologic grade is a strong prognostic factor in ST sarcomas [32]. High-grade sarcomas are at high risk of local recurrence and metastasis and are associated with poor prognosis [33,34,35,36]. In contrast, the prognosis of low-grade ST sarcoma is very good, with a metastasis-free survival probability at 5 years of more than 90% [37]. Therefore, DTPI may be very useful for the rapid detection of aggressive high-grade ST sarcomas. High [^18^F]FDG uptake and/or high RI in ST masses are very specific patterns for aggressive high-grade sarcoma. These metabolic features can alert the oncological team for optimal management, aid in choosing an appropriate part of the lesion to biopsy, reduce diagnostic delay, and enable optimal treatment for more aggressive ST sarcomas [38,39]. 

DTPI highlights the tumor burden activity and aggressiveness of ST sarcomas. Future studies are needed to evaluate the prognostic value of DTPI in ST sarcoma and its correlation with tumor recurrence and metastasis-free survival probability. In lung, breast, lymphoma, and head and neck cancer, DTPI was a strong predictor of event-free survival [40,41,42,43,44]. 

Despite the high specificity of DTPI, some benign lesions may show high FDG avidity and high RI. In our population of benign tumors, three lesions had a significant positive RI: two patients with tenosynovial giant cell tumors and one patient with steatonecrosis (Table 2). These ‘false-positive’ findings may be, at least partially, explained by the cellular composition of these lesions, which are rich in activated macrophages [45,46,47]. Similar to benign soft tissue lesions, DTPI also showed a significant RI in active tuberculous lesions or other granulomatous diseases due to activated macrophages [48,49]. Therefore, DTPI may not be helpful in tumors with an inflammatory-rich environment, such as tenosynovial giant cell tumors [50]. 

Our study has several limitations. First, patient recruitment was performed at the discretion of the multidisciplinary oncology team. Therefore, possible selective bias occurred by recruiting benign tumors with clinical and/or MRI criteria of local aggressiveness, such as tenosynovial giant cell tumors or desmoid tumors [51,52]. Moreover, some tumors with pure myxoid content at MRI were not evaluated by [^18^F]FDG PET/CT (no significant uptake). Second, we evaluated only the intensity of the metabolic activity (e.g., SUVmax) and the RI. We did not evaluate other [^18^F]FDG PET/CT parameters that correlate with tumor malignancy, such as metabolic tumor volume, total lesion glycolysis, and metabolic heterogeneity [53]. In this study, Chen et al. developed a regression model that included SUVmax and a metabolic heterogeneity factor. The RI may also appear as an interesting parameter to investigate in future multivariate analyses to better characterize ST tumors. Third, the results of our study need to be confirmed in a larger, more homogeneous population to better reflect the metabolic behavior of each tumor subtype. 

## 5. Conclusions

DTPI is a useful technique to differentiate malignant from benign ST tumors. High retention of [^18^F]FDG uptake during delayed acquisition is a very specific parameter for tumor malignancy. Moreover, this technique detected more aggressive malignancies (i.e., high-grade ST sarcomas) and may therefore be a useful tool to guide and optimize the management of an ST mass.

## Figures and Tables

**Figure 1 diagnostics-13-03202-f001:**
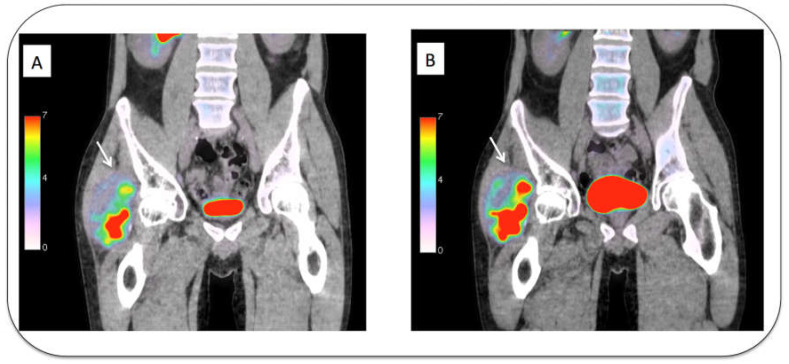
(**A**,**B**) Tumor in the soft tissue of the right buttock (arrow) with high [^18^F]FDG uptake at standard acquisition time (**A**, SUVmax = 16.9) and with a significant increase of [^18^F]FDG uptake during delayed acquisition (**B**, SUVmax = 23.4; RI = +39%). Pathological analysis confirmed a high-grade liposarcoma.

**Figure 2 diagnostics-13-03202-f002:**
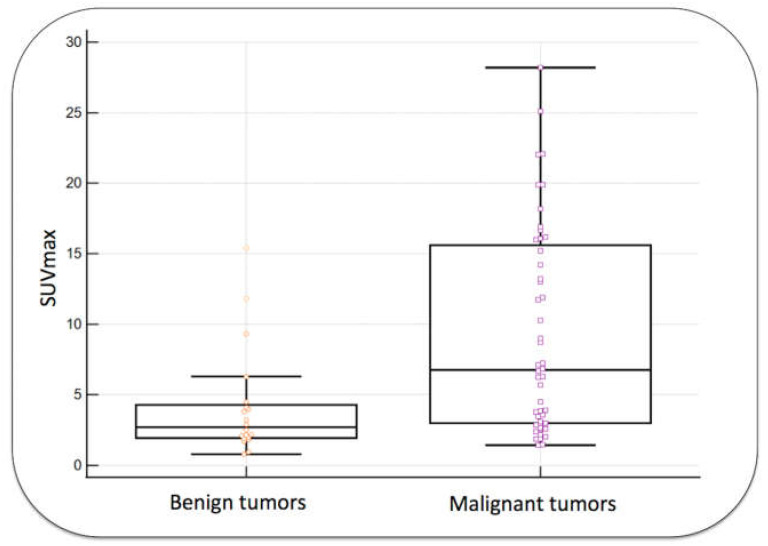
SUVmax in benign and malignant tumors. The groups were significantly different (*p*: 0.002). Orange and pink shapes show individual results for each tumor.

**Figure 3 diagnostics-13-03202-f003:**
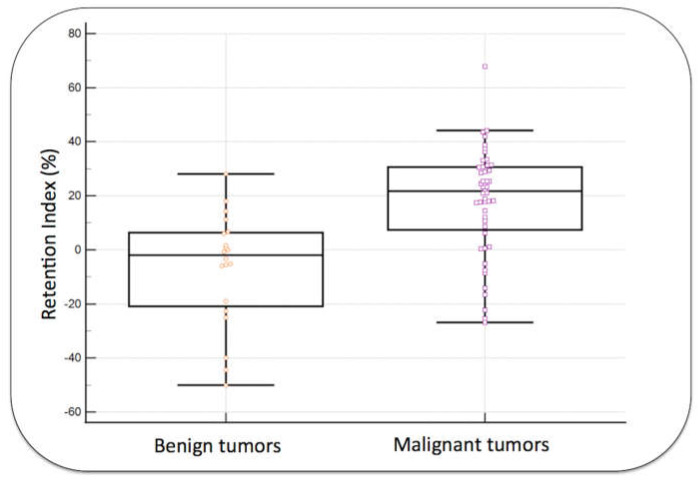
Retention index (RI) in benign and malignant tumors. The groups were significantly different (*p* < 0.0001).

**Figure 4 diagnostics-13-03202-f004:**
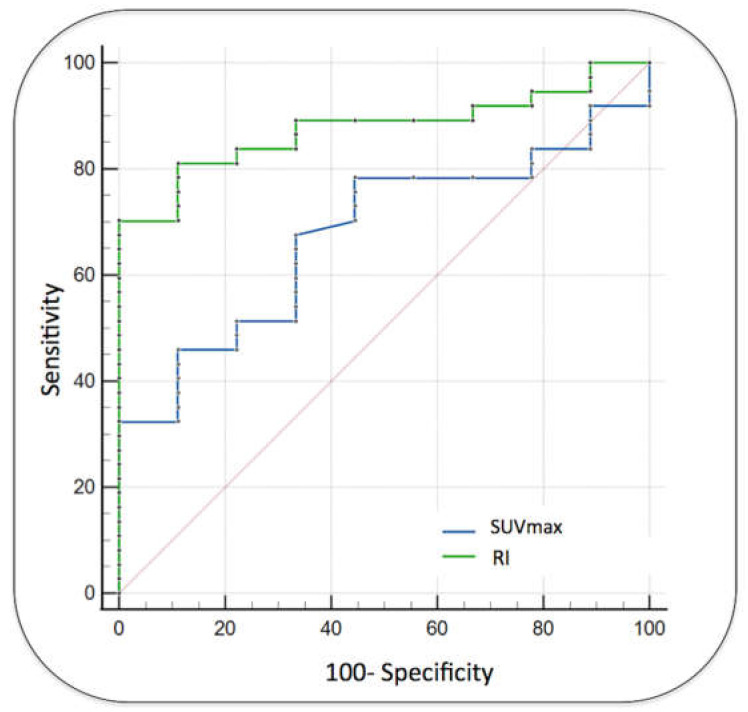
Diagnostic performance for tumors with at least mild metabolic activity at t1 (SUVmax ≥ 3). In this tumor population, an RI of >17.9% identified malignant tumors with an Sp of 100% and an Se of 70.3% (AUC = 0.88) and significantly outperformed SUVmax (AUC = 0.68, *p*: 0.01). The red line indicates the line of identity of the ROC curve.

**Table 1 diagnostics-13-03202-t001:** SUVmax and RI in the different malignancy subtypes.

Malignant Tumor Subtypes	Frequency (%)	SUVmax Median (IQR)	RI Median (IQR)
Liposarcoma	12 (25%)	4.9(11.2)	+18.3% (43.5%)
Undifferentiated high-grade sarcoma	10 (20.8%)	12.6 (13.2)	+20.5% (22.7%)
Myxofibrosarcoma	7 (14.6%)	3.9 (12.9)	+17.6% (30.7%)
Spindle cell sarcoma	3 (6.2%)	3.9 (10.6)	+30.8% (10.4%)
Ewing sarcoma of soft tissue	2 (4.2%)	5.2 (2.9)	+24.6% (13.2%)
Leiomyosarcoma	2 (4.2%)	11.6 (9)	+16% (2.8%)
Synovial sarcoma	1 (2%)	1.5	−26.8%
Angiosarcoma	1 (2%)	5.7	+24.6%
Clear cell sarcoma	1 (2%)	16	+67.8%
Follicular dendritic cell sarcoma	1 (2%)	22.1	+18.2%
Chondrosarcoma (ST metastasis)	1 (2%)	2.7	−14.2%
Osteosarcoma	1 (2%)	6.2	+33.5%
Ileal GIST	1 (2%)	4.5	+23.8%
Chordoma (ST pelvis metastasis)	1 (2%)	2.6	+6.2%
Lymphoma (marginal zone)	1 (2%)	3.6	+30.6%
Melanoma (metastasis)	1(2%)	13	+33.2%
Malignant pecoma	2 (4%)	17.1 (16.1)	+23.1% (4%)
Total	48 (100%)	6.8 (12.6)	+21.8% (23.3%)

IQR: interquartile range; ST: soft tissue; RI: retention index.

**Table 2 diagnostics-13-03202-t002:** SUVmax and RI in the different subtypes of benign ST tumors/lesions.

Benign Lesions/Tumors Subtypes	Frequency (%)	SUVmax Median (IQR)	RI Median (IQR)
Desmoid tumor	4 (20%)	5.1 (3.9)	+3.4% (15.5%)
Vascular malformation	4 (20%)	1.9 (0.4)	−5.6% (12.9%)
Myxoma	3 (15%)	2.9 (1.6)	−0.7% (29%)
Lipoma	2 (10%)	0.9 (0.2)	−47.2% (6%)
Tenosynovial giant cell tumor	2 (10%)	13.6 (3.6)	+16.1% (3.6%)
Hibernoma	1 (5%)	4.5	+11.1%
Granuloma	1 (5%)	2.1	−3.3%
Schwannoma	1 (5%)	3.2	−5.6%
Steatonecrosis	1 (5%)	2.1	+28.1%
Fibroma	1 (5%)	2.2	−22.7%
Total	20	2.9 (2.4)	−0.7% (31.5%)

IQR: interquartile range; RI: retention index.

**Table 3 diagnostics-13-03202-t003:** SUVmax and RI in low-grade and high-grade sarcomas.

	Low Grade (Grade 1–2)	High Grade (Grade 3)	*p*-Value
Frequency (%)	16 (40%)	24 (60%)	
SUVmax Median (IQR)	2.6 (1.8)	13.7 (9.9)	*p* < 0.001
RI Median (IQR)	+0.4% (22.8%)	+26.9% (14.5%)	*p* < 0.001

IQR: interquartile range.

## Data Availability

The data presented in this study are available on request from the corresponding author.

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
