# Peer review of "Diagnostic Superiority of Dual-Time Point [18F]FDG PET/CT to Differentiate Malignant from Benign Soft Tissue Tumors"

_diagnostics, 2023, doi:10.3390/diagnostics13203202_

Round 1
Reviewer 1 Report
The authors investigate whether dual-time point protocol (with retention index) improves the diagnostic performance of FDG PET in differentiating benign and malignant soft tissue neoplasm. Although malignant soft tissue tumor is not common, this topic is clinically relevant and may improve our clinical practice.
My comments and suggestions are as follows,
1. This study aimed to use dual-time point FDG PET to differentiate malignant and benign soft tissue tumors. Currently, histopathology is the standard tool to ascertain the diagnosis. The authors may need to describe some advantages of images over biopsy and some drawbacks of histopathology in these patients in the Introduction. So the clinical value of imaging to aid in differentiating benign and malignant soft tissue tumors can be highlighted.
2. Please provide the reconstruction parameters (such as filter and matrix sizes, TOF? or PSF?) for FDG PET in the materials and methods. Also, the authors mentioned they used the CT with a dose modulation protocol. What was the range of tube current?
3. What was the rationale for the arbitrary SUVmax cutoff of 3?
4. The authors present their SUVmax and RI with median and IQR. I think the distributions of SUVmax and RI were probably non-parametric. However, the authors used a t-test to examine the differences of SUVmax and RI between groups. I would suggest using a non-parametric test (U test). Please revise the results (Line 190-191 and Figures 2 and 3).
5. Line 166-167, the authors present the median of patients' age. Since median age was used, the authors should present IQR instead of a confidencen interval.
6. I would suggest the authors also present the data of delayed SUVmax. Did delayed SUVmax helpful in differentiating benign and malignant soft tissue tumors?
7. In this study, the dual-time point protocol is not helpful for soft tissue tumors with less FDG avidity. The authors may need to add a paragraph in the discussion to tell the readers that low FDG avid tumors may still need to be biopsied. So they won't miss possible low-grade cancers.
8. Benign tumors, such as giant cell tumors, may also show high SUV avidity and retention (Table 2). The authors may need to highlight this pitfall of the dual-time protocol in the Discussion.
9. The histopathologies of their patients were heterogeneous. Different soft tissue tumors may present with different SUV avidity. For example, melanoma may present with high avidity, while marginal zone lymphoma often presents low avidity. The authors may need to mention this in their limitations.
10. The authors cited a reference ( BMC Med Imaging. 2020 Jul 25;20(1):85.). This reference also evaluated the intensity, volumetric, and heterogeneity of FDG PET features in differentiating benign and malignant soft tissue tumors. This reference also used a multivariate regression model to improve the performance. I suggest the authors try regression to see if modeling is feasible and if the model performs better than individual features.
Author Response
Dear reviewer,
We would like to thank the reviewers for their comments and suggested additions to our manuscript. We herewith submit a revised version of our manuscript along with a reply letter addressing the comments of the reviewers. We hope to have addressed your concerns satisfactorily.
Comments reviewer 1
The authors investigate whether dual-time point protocol (with retention index) improves the diagnostic performance of FDG PET in differentiating benign and malignant soft tissue neoplasm. Although malignant soft tissue tumor is not common, this topic is clinically relevant and may improve our clinical practice.
My comments and suggestions are as follows,
- This study aimed to use dual-time point FDG PET to differentiate malignant and benign soft tissue tumors. Currently, histopathology is the standard tool to ascertain the diagnosis. The authors may need to describe some advantages of images over biopsy and some drawbacks of histopathology in these patients in the Introduction. So the clinical value of imaging to aid in differentiating benign and malignant soft tissue tumors can be highlighted.
Answer (written as well in the manuscript):
The high diversity, rarity, intrinsic complexity and intratumoral histological heterogeneity of soft tissue sarcomas pose significant challenges to confirm the pathological diagnosis, with inaccuracies reaching up to 30%. .
« Furthermore, due to the similar appearance of benign and malignant soft tissue tumors,the risk of diagnostic error is high. In a series of 581 lesions secondarly reviewed by an expert pathologist, major diagnostic errors affecting patient management were found in 148 cases (25%) with 20 benign lesions being reclassified as malignant (Rupani et al. Diagnostic differences in expert second opinion cases at a tertiary sarcoma cente. Sarcoma.2020). To optimize tumor characterisation, combining additional analyses such as imaging with pathological diagnosis may be beneficial. This has been illustrated by Kuhn et al. who showed that one should ensure there is a concordance between the imaging and the pathologic diagnosis especially in the case of diagnostic dilemmas (Kuhn et al. Soft tissue pathology for the radiologist : a tumor board primer with 2020 WHO classification update. Skeletal radiology 2021). Imaging allows to assess non-invasively the intra-lesion heterogeneity and may guide biopsy towards the most metabolically active regions, hence leading to more representative biopsies. »
- Please provide the reconstruction parameters (such as filter and matrix sizes, TOF? or PSF?) for FDG PET in the materials and methods. Also, the authors mentioned they used the CT with a dose modulation protocol. What was the range of tube current?
Answer (written as well in the manuscript):
All images were acquired on a TOF-PET/CT (Philips Gemini TF64) with a time-resolution of 600ps. standard FDG iterative reconstruction (OSEM) was used, i.e. 4x4x4 mm3 reconstructed voxel size (matrix 144 x144) with 3 iterations x 33 subsets. CT images were acquired with standard parameters i.e. tube voltage 120 kV and effective tube current up to 100 mA.
- What was the rationale for the arbitrary SUVmax cutoff of 3? In our population of tumors, we tested different SUVmax cutoffs. Only the SUVmax cutoff of 3 was significant separating well the ROC curves in figure 4. The manuscript was changed as follows :
Answer (written as well in the manuscript):
When including all patients, we did not observe any significant differences in AUC’s between SUVmax and RI. Since this comprises a wide spectrum of metabolic activities, we tested different SUVmax cut-offs by comparing the ROC curves and the cut-off SUVmax ³ 3 yielded the most discriminant findings. Hence, we performed a secondary analysis including only tumors with a SUVmax ³ 3 and this showed
significant differences in AUC’s between SUVmax to RI
- The authors present their SUVmax and RI with median and IQR. I think the distributions of SUVmax and RI were probably non-parametric. However, the authors used a t-test to examine the differences of SUVmax and RI between groups. I would suggest using a non-parametric test (U test). Please revise the results (Line 190-191 and Figures 2 and 3).
Answer (written as well in the manuscript):
We fully agree with the reviewer. The SUVmax and RI values were not normally distributed and as such, a non-parametric test is prefered. We performed a new analysis using the Mann-Whitney U test as sugegsted by the reviewer. P values remained statistically significant when using SUVmax (p = 0.002) and RI (p< 0.0001) to discriminate malignant from benign lesions.
- Line 166-167, the authors present the median of patients' age. Since median age was used, the authors should present IQR instead of a confidencen interval.
Answer (written as well in the manuscript)::
We agree that it is prefered to report the IQR when using a median value. The emdian age was 62.5 years with an IQR of 24. This has been corrected in the manuscript. « 68 patients were included in this study: 34 males and 34 females with a median age of 62.5 years (IQR: 24). »
- I would suggest the authors also present the data of delayed SUVmax. Did delayed SUVmax helpful in differentiating benign and malignant soft tissue tumors?
Answer (written as well in the manuscript):
We indeed looked at the peformance of the delayed SUVmax value, but it showed a similar performance to to SUVmax (t1) with a cutoff > 5.1 (Se=60.4%, Sp=85%). ROC curves were not significantly different (AUC= 0.76 [CI 95%: 0.64; 0.86], p=0.06).
- In this study, the dual-time point protocol is not helpful for soft tissue tumors with less FDG avidity. The authors may need to add a paragraph in the discussion to tell the readers that low FDG avid tumors may still need to be biopsied. So they won't miss possible low-grade cancers.
Answer :
In our data, DTPI was very efficient for distinguishing malignant tumors especially those with at least mild initial [18F]FDG uptake (Figure 3, specificity reaching 100%). Nevertheless, this delayed PET acquisition is useless for tumors wih with low [18F]FDG uptake (tumors with SUVmax < 3 in our study). « Therefore, DTPI cannot help to differentiate low-grade saromas from benign lesions, both of which often show mild [18F]FDG uptake. » Independent of the metabolic activity, all lesions should still be biopsied to obtain a final histopathologic diagnosis.
- Benign tumors, such as giant cell tumors, may also show high SUV avidity and retention (Table 2). The authors may need to highlight this pitfall of the dual-time protocol in the Discussion.
Answer (written as well in the manuscript):
We acknowledge that some subtypes of benign lesions may show high FDG avidity. In our population of benign tumors, three lesions had a significant postive RI : two patients with tenosynovial giant cell tumors and one patient with steatonecrosis (Table 2). These ‘false positive’ findings may be, at least partially, explained by the cellular composition of these lesions, which are rich in activated macrophages (Lewis et al. Hot shoulder PEY/CT lesion : unusual presentation of tenosunovial giant cell tumor, radiology case reports 2018 ; Pallas et al. Intense FDG uptake in an intra-articular giant cell tumor of the tendon sheath. Radiology cases reports 2009.Burkholz et al. posttraumatic pseudolipoma (fat necrosis) mimicking atypical lipoma or liposacoma on MRI, RAdiol Cas Rep. 2007). Similarly to benign soft tissue lesions, DTPI also showed significant RI in active tuberculous lesions or other granulomatous diseases due to activated macrophages (Wei-Ye Yu. Uptades on FDG PET/CT as a clinical tool for tuberculosis evaluation and therapeutic monitoring QIMS 2019 ; Huang et al. Dual time point fdg pet/ct in the diagnosis of siladary pulmonary lesions in a region with endemic granulomayous diseases, ann nucl med 2016). Therefore, DTPI may not be helpful in tumors with an inflammatory rich environment such as tenosynivial giant cell tumors (Robert et al. Update on tenosynovial giant cell tumor, an inflammatory arthritis with neoplastic features).
- The histopathologies of their patients were heterogeneous. Different soft tissue tumors may present with different SUV avidity. For example, melanoma may present with high avidity, while marginal zone lymphoma often presents low avidity. The authors may need to mention this in their limitations.
Answer (written as well in the manuscript):
We agree that we have a heterogeneous tumor composition, which is inherently linked to the wide variety of soft tissue sarcomas, a notable feature of this tumer type. Consequently, the tumors showed a
variable biological behavior and metabolic activity (Sbaraglia et al. The 2020 WHO classification of soft tissue tumours Pathologica. 2021).. Future investigations in a larger, and more homogeneous, population would be interesting to better characterize the metabolic behavior of the different subtypes of ST tumors.
- The authors cited a reference ( BMC Med Imaging. 2020 Jul 25;20(1):85.). This reference also evaluated the intensity, volumetric, and heterogeneity of FDG PET features in differentiating benign and malignant soft tissue tumors. This reference also used a multivariate regression model to improve the performance. I suggest the authors try regression to see if modeling is feasible and if the model performs better than individual features.
Answer : We made reference to this article in the manuscript. Indeed, the group of Chen et al. evaluated 5 imaging parameters to characterize the malignant nature of soft tissue tumors : tumor size, SUVmax, MTV, TLG and heterogeneity factor (HF). Therefore, a multivariate analysis with a regression model can be very useful to highlight the best discriminative parameters. However, we only investigated two distinct metabolic parameters, obviating the interest to perform a multivariate regression model.. This sugestion has been incuded in the manuscript as follows :
Second, we evaluated only the intensity of the metabolic activity (e.g. SUVmax) and the RI. We did not evaluate other [18F]FDG PET/CT parameters such as metabolic tumor volume, total lesion glycolysis and metabolic heterogeneity (4). « In this study, Chen et al. developed a regression model that included SUVmax and a metabolic heterogeneity factor. RI may also appear as an interesting parameter to investigate in future multivariate analyses to better characterize ST tumors. »
Reviewer 2 Report
1. line 72: mixoid change into myxoid
2. line 107: growth.=> deleted dot
3. Table 1 Total SUVmax 6.8 (12.6), meaning?
Total RI +21.8% (23.3%), meaning?
4. Table 2 Total SUVmax 2.9 (2.4), meaning?
Total RI -0.7% (31.5%), meaning?
5. Figure 4 line 258: add Delta SUVmax=RI.
6. Line 274, 277, 281: signal change into uptake
Author Response
Dear reviewer,
We thank you very much for your review.
Comments reviewer 2
- line 72: mixoid change into myxoid
answer : done
- line 107: growth.=> deleted dot
answer : done
- Table 1 Total SUVmax 6.8 (12.6), meaning? Total RI +21.8% (23.3%), meaning?
Answer : for all tumors (total), the median SUV max and RI and IQR in brackets. It is as well explained in results line 190.
- Table 2 Total SUVmax 2.9 (2.4), meaning Total RI -0.7% (31.5%), meaning?
Answer : for all tumors (total), the median SUV max and RI and IQR in brackets. It is as well explained in the results line 190.
- Figure 4 line 258: add Delta SUVmax=RI.
Answer :done
- Line 274, 277, 281: signal change into uptake
Answer :done
Reviewer 3 Report
Comments and Suggestions for Authors
Τhe manuscript deals with a very interesting object that appears a great clinical value. However, that are some issues that should be taken into account by authors. A few specific suggests are given for the author's consideration to make the article more comprehensive.
1. Inter and intra observer variability should be considered in order the objectivity to be improved
2. The utilization of a significant higher number of malignant vs benign cases might affect the balance and the final selection of the correct characterization threshold / score.
3. The large number of disease categories (Table 2-3) together with the limited number of cases per specific pathology, leads to poor statistics.
4. The differentiation threshold between malignant/benign, the numerical value, might be a PET/CT system sensitive parameter or could be a unique factor?
Author Response
Dear reviewer,
We thank you very much for your review.
Comments and Suggestions reviewer 3
- Inter and intra observer variability should be considered in order the objectivity to be improved
Answer : Our study was analyzed by only a single nuclear medicine physician using a MIM software. We defined a volume of interest (VOI) around the tumor and the software automatically generated the SUVmax of the lesion. Thereofore, no inter/ intra variability is expected. Furthermore, SUVmax is a robust semi- quantitative parameter parameter known to be highly reproducible (examples of studies 10.2967/jnumed.107.050187 ; https://doi.org/10.2967/jnumed.108.054239).
The following text has been added in the manuscript : Analyses were performed using MIM software, version 7.1.3 (MIM Software Inc., Cleveland, OH). « With this software, we automatically defined the SUVmax of the tumor using a volume of interest. »
- The utilization of a significant higher number of malignant vs benign cases might affect the balance and the final selection of the correct characterization threshold / score.
Answer :We agree that this may affect the final cut-off value but we prospectively included all consecutive cases, so there is no inclusion bias. Nevertheless, a possible selection occurred by recruiting only tumors with clinical or MRI criteria of aggressivness. This limitation was mentionned in the manuscript :
« the patient recruitment has been carried out at the discretion of the multidisciplinary oncology team. Therefore, possible selective bias occurred by recruiting benign tumors with clinical and/ or MRI criteria of local aggressiveness such as tenosynovial giant cell tumors or desmoid tumors »
- The large number of disease categories (Table 2-3) together with the limited number of cases per specific pathology, leads to poor statistics.
Answer : Indeed, we have few cases per pathology, it is inherently related to the great variety and heterogenity of soft tissue tumors. Nevertheless, It is interesting to have the results in each pathology, not for a statistical purpose but to highlight the false negative results (example : synovial sarcoma) and the false postive results (tenosynovial giant cell tumors). We have discussed false negative and false postive cases in the discussion section.
We also added the following text at the end of the manuscript :
« Third, the results of our study need to be confirmed in a larger more homogeneous population to better reflect the metabolic behavior of each tumor subtype. »
- The differentiation threshold between malignant/benign, the numerical value, might be a PET/CT system sensitive parameter or could be a unique factor?
Answer : We agree that the SUVmax is highly dependent on several factors including imaging equipment, reconstruction parameters, incubation time…. Therefore, an SUVmax cut-off cannot be blindly extrapolated to other centers. However, the aim of our study was to highlight the retention index, RI. This parameter should be less affected by these factors because it reflects the change in SUVmax within one individual lesion at two time points using the same PET/CT system. We have added this text in the methods section :
A second delayed acquisition (t2) was obtained on the tumor site at 180 minutes PI, with a longer PET acquisition (3.5 minutes per bed position), aimed at reducing noise caused by fluorine-18 decay. « This second acquisition was performed on the same PET/CT system to avoid SUV variability due to the equipment. »
Round 2
Reviewer 1 Report
All my suggestions and comments have been well addressed.